# A case report describing the immune response of an infant with congenital heart disease and severe COVID-19

Danielle Wurzel[1,2,3,4,9], Melanie R. Neeland [1,2,9], Jeremy Anderson[1,9], Yara-Natalie Abo[1,2,3], Lien Anh Ha Do [1,2], Celeste M. Donato[1,2], Julie E. Bines[1,2,3], Zheng Quan Toh [1,2], Rachel A. Higgins[1], Sedi Jalali[1], Theresa Cole[1,2,3], Kanta Subbarao [2,5], Alissa McMinn[1], Kate Dohle[1], Gabrielle M. Haeusler[1,2,3], Sarah McNab[1,2,3], Annette Alafaci[1], Isabella Overmars [1], Vanessa Clifford[1,2,3], Lai-yang Lee[1,2,3], Andrew J. Daley[1,2,3], Jim Buttery[1,2,3,6], Penelope A. Bryant[1,2,3], David Burgner [1,2,3], Andrew Steer[1,2,3], Shidan Tosif[1,2,3], Igor E. Konstantinov[2,3,8], Trevor Duke[1,2,3,10], Paul V. Licciardi [1,2,10✉], Daniel G. Pellicci [1,2,7,8,10✉] & Nigel W. Crawford[1,2,3,10]

## Abstract

**Background** Children with SARS-CoV-2 infection generally present with milder symptoms or are asymptomatic in comparison with adults, however severe disease occurs in a subset of children. To date, the immune correlates of severe COVID-19 in young children have been poorly characterised.

**Methods** We report the kinetics of immune responses in relation to clinical and virological features in an infant with acute severe COVID-19 using high-dimensional flow cytometry and multiplex cytokine analysis.

**Results** Systemic cellular and cytokine profiling show an initial increase in neutrophils and monocytes with depletion of lymphoid cell populations (particularly CD8 + T and NK cells) and elevated inflammatory cytokines. Expansion of memory CD4 + T (but not CD8 + T) cells occurred over time, with a predominant Th2 bias. Marked activation of T cell populations observed during the acute infection gradually resolved as the child recovered. Substantial in vitro activation of T-cell populations and robust cytokine production, in response to inactivated SARS-CoV-2 stimulation, was observed 3 months after infection indicating durable, long-lived cellular immune memory.

**Conclusions** These findings provide important insights into the immune response of a young infant with severe COVID-19 and will help to inform future research into therapeutic targets for high-risk groups.

## Plain language summary

SARS-CoV-2 infection can cause COVID-19, which is usually a mild disease in children. However, severe illness requiring intensive care management can occur, particularly in younger children and those with chronic disease. The immune system likely plays an important role in susceptibility to severe disease, but few studies have examined the immune response in infants with severe COVID-19. Here, we provide an in-depth analysis of the clinical features and immune response over time in a 5-month-old infant with congenital heart disease and severe COVID-19. Robust immune responses were observed up to 3 months following infection, providing evidence of durable and long-lived immunity against SARS-CoV-2. These findings provide important insights into the immune responses of a young infant with severe COVID-19 and might help inform future research into therapeutic targets.

[1] Infection and Immunity, Murdoch Children's Research Institute, Melbourne, VIC, Australia. [2] Department of Paediatrics, University of Melbourne, Melbourne, VIC, Australia. [3] Royal Children's Hospital, Melbourne, VIC, Australia. [4] Melbourne School of Population and Global Health, The University of Melbourne, Melbourne, VIC, Australia. [5] WHO Collaborating Centre for Reference and Research on Influenza, Peter Doherty Institute for Infection and Immunity, Melbourne, VIC, Australia. [6] Monash Children's Hospital, Melbourne, VIC, Australia. [7] Department of Microbiology and Immunology, University of Melbourne, Melbourne, VIC, Australia. [8] Melbourne Centre for Cardiovascular Genomics and Regenerative Medicine, Murdoch Children's Research Institute, Melbourne, VIC, Australia. [9] These authors contributed equally: Danielle Wurzel, Melanie R. Neeland, Jeremy Anderson. [10] These authors jointly supervised this work: Trevor Duke, Paul V. Licciardi, Daniel G. Pellicci, Nigel W. Crawford. ✉email: paul.licciardi@mcri.edu.au; dan.pellicci@mcri.edu.au

Children have milder coronavirus disease-2019 (COVID-19) than adults; however, younger children (<4 years of age) may develop more severe disease, with higher hospitalisation and intensive care unit admission, than older children and young adults[1]. Children with underlying chronic conditions, such as congenital cardiac anomalies are at increased risk; however, data on the clinical, virological, and immunological features of infants with severe COVID-19 are scarce.

The present study describes the clinical course of an infant with a chronic heart condition admitted to the Royal Children's Hospital in Melbourne with severe COVID-19. Virological and immunological responses show a high viral load in naso-/oropharyngeal specimens, elevated SARS-CoV-2 IgG and neutralising antibody responses, robust activation of CD4 + T cell subsets, and cytokine production. A robust cellular immune memory response following stimulation with inactivated SARS-CoV-2 is also a feature of severe COVID-19 in this infant. This data increases our understanding of severe COVID-19 in high-risk infants and will be important for the development of therapeutics.

## Methods

**Clinical**. This child was recruited as part of the "COVID-19 at the Melbourne Children's Campus: Program of Research". The research team was notified by the clinical laboratory at The Royal Children's Hospital, Melbourne of all SARS-CoV-2 positive samples (as detected by RT-PCR on oro-/nasopharyngeal samples +/− serology). Written informed consent was obtained electronically from the child's parents/guardians. Longitudinal clinical data and bio-specimen collection (including blood, stool, urine, and oro-/nasopharyngeal samples) were undertaken at designated time-points during the acute illness and convalescent period as per the study protocol. Bio-specimens were processed immediately for whole blood flow cytometry and viral molecular testing, additional samples were stored in a designated bio-bank at the MCRI research facility for later processing.

Informed consent was obtained from the child's parent/guardian (via e-consent) prior to the child's inclusion in the study and consent was provided to publish the clinical details of the child reported in this study. The study received ethics approval from The Royal Children's Hospital (RCH), Melbourne, Human Research and Ethics Committee (HREC 63103). The healthy control sample was obtained following informed consent from the Melbourne Children's Heart Tissue Bank (MCHTB) under ethics approval by the RCH HREC (#38192).

**Virology**. Combined oropharyngeal and deep nasal swabs were collected according to national guidelines using dry FLOQSwabs® (Copan, Brescia, Italy). FLOQSwabs were eluted in 500 μL of phosphate-buffered saline (PBS). Nucleic acid extraction was performed using 200 μL eluent, extracted using the Roche MagNA Pure 96 extraction system (Roche, Basel, Switzerland), according to the manufacturer's instructions. 10 μL extracts were tested using the LightMix® Modular SARS and Wuhan CoV E-gene kit (TIB Molbiol, Berlin, Germany) on the Roche Light-Cycler 480 II Real-Time PCR System. Positive tests were confirmed using the AusDiagnostics Respiratory Pathogens 16-well assay (AusDiagnostics, Mascot, Australia) system (targeting ORF-1 and ORF-8 genes), on the AusDiagnostics High-Plex 24.

RNA was manually extracted from 280 μL of urine and 140 μL of 20% (w/v) faecal suspension[2] and then eluted in 60 μL sterile, molecular water (Life technologies, Australia), using the QIAamp viral RNA kit (QIAgen GmbH, Hilden, Germany) according to the manufacturer's instructions. RNA was extracted from 200 μL whole blood and eluted in 100 μL of elution buffer using the MagNA Pure 96 DNA and Viral RNA Small Volume Kit (Roche)

according to the manufacturer's instructions. Three sets of previously published RT-PCR primers were used; one targeting the E-gene and two targeting the RdRp gene, were used on an ABI 7500[3, 4]. The SARS-CoV-2 standard (Exact Diagnostic, US) was used as positive control for the PCR. Stool collected at four timepoints (days 3, 5, 10, and 28) were negative for the presence of SARS-CoV-2 RNA. The urine collected on day 3 was positive for the presence of SARS-CoV-2 RNA, with the subsequent samples collected on days 5 and 10 negative (Supplementary Table 2).

## Immunology

*Flow cytometry of whole blood samples*. Blood was collected in EDTA tubes and 100 μL of whole blood was aliquoted for flow cytometry analysis. Whole blood was lysed with 1 mL of red cell lysis buffer for 10 min at room temperature. Cells were washed with 1 mL PBS and centrifuged at $350 \times g$ for 5 m. Following two more washes, cells were resuspended in PBS for viability staining using near infra-red viability dye according to the manufacturer's instructions. The viability dye reaction was stopped by the addition of FACS buffer (2% heat-inactivated FCS in 2 mM EDTA) and cells were centrifuged at $350 \times g$ for 5 min. Cells were then resuspended in human FC-block according to manufacturer's instructions for 5 min at room temperature. The whole blood cocktail (Supplementary Table 3) made up at 2X concentration were added 1:1 with the cells and incubated for 30 min on ice. Following staining, cells were washed with 2 mL FACS buffer and centrifuged at $350 \times g$ for 5 m. Cells were then resuspended in 2% PFA for a 20 min fixation on ice, washed, and resuspended in 150 μL FACS buffer for acquisition using the BD LSR X-20 Fortessa and BD FACS DIVA V 9.0 software. For all flow cytometry experiments, compensation was done at the time of sample acquisition using compensation beads. Supplementary Figure 3 depicts the manual gating strategy for whole blood samples.

**Flow cytometry of peripheral blood mononuclear cells**. Cryopreserved peripheral blood mononuclear cells (PBMCs) were thawed at 37 °C then washed with 10 mL $R_{10}$ media (RPMI-1640 medium supplemented with 10% fetal bovine serum, 200 nM L-glutamine, 1000IU penicillin-streptomycin) and centrifuged at $400 \times g$ for 5 min. PBMCs were washed with 5 mL PBS and centrifuged at $400 \times g$ for 5 min then blocked (50 μl of 1% human FC-block and 10% normal rat serum in PBS) for 15 min on ice. PBMCs were washed with 1 mL FACS buffer and centrifuged at $400 \times g$ for 5 min then stained with 50 μl PBMC cocktail (Supplementary Table 4) for 20 min on ice. PBMCs were washed then resuspended in 4% PFA for a 10 min fixation on ice, washed, and resuspended in 100 μl FACS buffer for acquisition using the Cytek Aurora. Compensation was performed at the time of acquisition using compensation beads. Supplementary Figure 4 depicts the manual gating strategy for PBMC samples.

**PBMC stimulation with inactivated SARS-CoV-2**. Cryopreserved peripheral blood mononuclear cells (PBMCs) were thawed at 37 °C then washed with 10 mL $R_{10}$ media. Cell viability was measured via trypan blue exclusion and $1 \times 10^6$ cells/mL were left unstimulated (control) or stimulated with inactivated SARS-CoV-2 (1:10 dilution) for 4 days at 37 °C, 5% $CO_2$. Following stimulation, supernatants were harvested to measure cytokines, chemokines, and growth factors, and PBMCs were stained for flow cytometry.

**Cytokine, chemokine, and growth factor measurements**. A commercial multiplex bead array kit (27-plex human cytokine

assay; Bio-Rad, New South Wales, Australia) was used to measure IL-1β, IL-1ra, IL-2, IL-4, IL-5, IL-6, IL-7, IL-8, IL-9, IL-10, IL-12(p70), IL-13, IL-15, IL-17A, eotaxin, FGF-basic, G-CSF, GM-CSF, MCP-1, IFN-γ, TNF-α, IP-10, RANTES, MIP-1α, MIP-1β, PDGF and VEGF according to manufacturer's instructions. Results were analysed on a Luminex 200 instrument (Luminex, Texas, USA) fitted with the Bio-Plex Manager Version 6 software, and results were reported in pg/mL.

IL-18 was measured by a commercial ELISA kit (R&D Systems, Minnesota, United States) according to the manufacturer's instructions. Results were measured at an optical density of 450 nm (reference wavelength 630 nm), and concentrations in pg/mL were derived from the standard curve.

## Results and discussion
**Clinical outcomes.** A 5-month old male infant was transferred to The Royal Children's Hospital (RCH, Melbourne, Australia) with an acute respiratory illness early in the COVID-19 pandemic. He had cyanotic congenital heart disease with single left ventricle physiology (congenitally corrected transposition of the great arteries, hypoplastic left aortic arch, and large patent ductus arteriosus) for which he had undergone partial correction with insertion of a systemic-to-pulmonary shunt and PDA closure. A day prior to presentation he developed low-grade fevers, cough,

and increased work of breathing (Fig. 1A, Day−2). At his local hospital, he was hypoxic (initial oxygen saturation was 70%) and was commenced on low-flow nasal prong oxygen. His echo showed good ventricular function. Reverse-transcriptase-poly-merase-chain-reaction (RT-PCR) for SARS-CoV-2 from a naso-pharyngeal/oropharyngeal swab was positive. Both parents and a sibling had been diagnosed with COVID-19 in the preceding week, but did not require hospitalisation.

The following day his respiratory status worsened and he developed severe respiratory and metabolic acidosis with pH 6.99, $PCO_2$ 55 mmHg, and lactate 13 mmol/L. He was intubated and ventilated and transferred to the intensive care unit for further management (Fig. 1A, Day 0). He required mechanical ventilation with pressure support of 17cmH$_2$0 and positive end-expiratory pressure of 8cmH$_2$0, respiratory rate of 25 breaths per minute and fraction inspired oxygen of 40%. There was evidence of haemodynamic compromise with relative bradycardia (heart rate 100 beats per minute), hypothermia (33°C), and hypotension (mean arterial blood pressure 35 mmHg). He was commenced on an infusion of adrenaline 0.05 mcg/kg/min. A plain chest film showed peri-hilar interstitial and ground-glass opacities with left lower lobe collapse and small bilateral pleural effusions (Fig. 1B). Notable initial laboratory investigations included elevated ferritin (9487 µg/L, normal range [NR] 11-87)

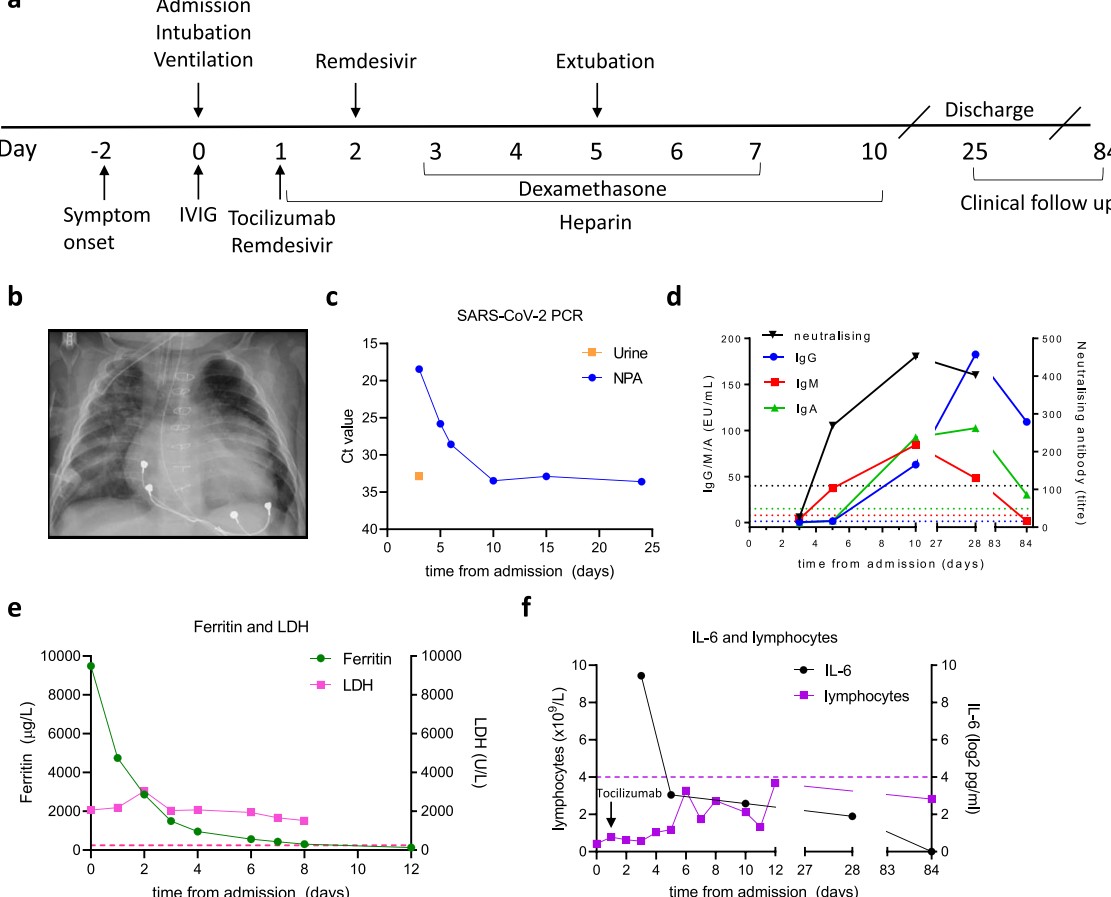

**Fig. 1 Clinical and laboratory characteristics of an infant with severe COVID-19. A** Clinical timeline showing immuno-modulatory therapies in relation to clinical course. **B** Plain chest film performed on admission illustrating bilateral interstitial infiltrates with left lower lobe collapse and consolidation. **C** Virologic findings illustrating SARS-CoV-2 Cycle threshold Ct values in naso-/oropharyngeal and urine specimens that reduce in association with clinical improvement. **D** Serum IgG, IgM, and IgA to S1 using in-house ELISA modified from Mount Sinai Laboratories, USA; and microneutralisation assay demonstrating a rapid rise in neutralising antibody titres. **E** Elevated ferritin and LDH with initial hyper-inflammatory picture and rapid reduction in ferritin coinciding with clinical improvement. **F** Persistent lymphopaenia; initial high IL-6 with reduction after administration of tocilizumab. IVIG intravenous immunoglobulin, LDH lactate dehydrogenase, NPA nasopharyngeal aspirate, EU ELISA units, Ct cycle threshold.

and d-dimers (5.86 µg/mL, NR < 0.5) with reduced lymphocytes (0.43x10e9/L, NR 4.0−10.0) (Supplementary Table 1).

He was treated with broad-spectrum anti-microbials, intravenous immunoglobulin (1 g/kg), and remdesivir (loading dose 5 mg/kg then 2.5 mg/kg then ceased as lyophilised solution was unavailable). Based on his cardiorespiratory deterioration and elevated inflammatory markers, therapies targeting the COVID-19 inflammatory response were administered, including tocilizumab 8 mg/kg and dexamethasone 0.15 mg/kg (twice daily for 5 days, Fig. 1A). Anti-coagulation with low-molecular-weight heparin was commenced to mitigate the occlusion of his cardiac shunt.

**Virological outcomes.** Cycle threshold (Ct) values for SARS-CoV-2 in upper airway swabs were initially low (indicative of high viral load) with detectable viral RNA present until day 25. SARS-CoV-2 RT-PCR was positive in urine at day 3 of admission (Fig. 1C); stool was negative at all time-points.

**Immunological outcomes.** Serology for SARS-CoV-2 S1 IgG, IgM, and IgA was initially negative (day 3) with seroconversion occurring by day 5 and IgG and IgA remaining positive until day 84 (Fig. 1D) indicating durable humoral immunity. Neutralising antibodies showed an early rise (days 3−5) with persistently high titres from day 10 through 28 (Fig. 1D). A rapid decline in inflammatory markers (e.g., ferritin and IL-6, Fig. 1E, F) was observed in association with clinical improvement and extubation occurred on day 5. Lymphocyte counts remained below normal reference ranges throughout admission and follow-up (Fig. 1F), a pattern also observed in adults with severe COVID-19 [5].

Flow cytometry was performed on whole blood collected 3, 5, 10, 28, and 84 days following admission (Fig. 2). Neutrophil and eosinophil proportions remained stable over the first 10 days, although a substantial increase in both cell types was observed at day 28, with a 3.5-fold increase in neutrophils and a 12.8-fold increase in eosinophils (Fig. 2A). Similar to reports from adults with severe COVID-19[5–7], cells of lymphoid origin, including CD4 + T cells, CD8 + T cells, B cells, and natural killer (NK) cells were markedly reduced at day 10, subsequently increasing (Fig. 2A). Classical (CD14+CD16−) monocytes changed substantially over time, with marked increases observed at days 5, 10, 28, and 84 days after admission. Intermediate (CD14+CD16+) monocytes were virtually absent at day 3, increasing at days 5 and 10, with a decline observed by day 28 and returning by day 84. Non-classical (CD14lowCD16+) monocytes were essentially absent at the first three time points, returning to the circulation by day 84 (Fig. 2A).

Unsupervised clustering analysis on whole blood flow cytometry data using FlowSOM[8] and UMAP[9] was also performed. The frequency of clusters at each time point revealed identical sequential changes to that observed by manual gating and identified the presence of a CD16low immature granulocyte cluster at day 3 and day 5 (Fig. 2B and Supplementary Fig. 1). PBMCs were collected at each timepoint and used for high-dimensional flow cytometric analysis of T cell subsets. For comparison, PBMCs from an age/sex-matched infant who had undergone correction for a tetralogy of Fallot was obtained at a single time-point. γδTCR+Vδ2+ T-cells, which were much lower at day 3 compared to the control, increased more than ten-fold (0.22% at day 3 to 2.83% at day 84) (Fig. 2C). Similarly, mucosal-associated invariant T-cell (MAIT) frequencies had increased (0.2−0.35% from day 3 to day 84 respectively, Fig. 2C). The proportion of Th2 cells, which were very low in the control child (2.36%, consistent with the previous reports[10]), increased in

the COVID-patient from 9.32% at day 3 to 28.2% at day 84, while the proportion of Th1, Th17, or Treg populations varied little over time (Fig. 2D). Effector and central memory CD4+ T-cell subsets expanded from day 3 onwards, however, this was not seen for CD8+ T-cells (Fig. 2D). CD69, a marker used to define activated T cells, was minimally expressed on the control PBMC sample (Fig. 2E). In contrast, high CD69 expression was observed on CD4+ T cells, CD8+ T cells, γδTCR+Vδ2+ T-cells, and MAIT cells at day 3 (Fig. 2E). CD69 expression on these T-cell subsets gradually declined after day 3 to levels akin to the control, although CD69+ expression on MAIT cells increased again by day 84 (Fig. 2E).

Multiplex analysis of serum samples revealed a broad array of cytokines produced throughout the course of infection (Fig. 2F). The acute phase was dominated by inflammatory cytokines such as IL-6, IL-8, and TNFα and the chemokine IP-10 (CXCL10) at day 3 (Fig. 2F), consistent with previous data from severe COVID-19 disease in adults[6, 7, 11, 12]. In addition, IFNγ, G-CSF, Eotaxin, MIP-1α, and MCP-1 were also highly elevated at this time-point compared to later time-points (Fig. 2F), indicative of a potent pro-inflammatory immune response and enhanced cell migration. The level of IL-6 was markedly reduced by day 5 after tocilizumab (anti-IL-6) therapy and remained low at subsequent time-points (Fig. 2F). Furthermore, early treatment with remdesivir, tocilizumab, and dexamethasone (and/or natural history of COVID-19 infection) correlated with a shift towards Th2-type anti-inflammatory cytokines (e.g IL-4, IL-5, IL-10, IL-13) by day 10. Notably, dexamethasone has a broad range of immunomodulatory effects, including downregulating B- and T-cell receptor signalling, reduced pro-inflammatory cytokine secretion (e.g., IL-1, TNFα, IL-6), and increasing anti-inflammatory IL-10 cytokine, consistent with the patterns observed in our study[13]. High levels of PDGF-BB, IL-1β, IL-2, IL-7, IL-15, and IL-17 were also observed at day 10 coinciding with disease resolution (Fig. 2F). By day 28, high levels of IL-1β and IL-15 were observed, while day 84 was characterised by increased IL-9, MIP-1β, and RANTES (Regulated on Activation, Normal T cell expressed and Secreted, Fig. 2F).

Interestingly, we also found IL-18 was highest on day 3 (Fig. 2F) which correlated with the greatest CD69 activation on both Vδ2+ γδ T cells and MAIT cells (Fig. 2E). Moreover, whilst IL-12 was predominantly detected at day 10, both IL-18 and IL-12 were detected at day 84 (Fig. 2F) correlating with high CD69 expression on MAIT cells at this time-point. Innate-like T-cells, such as Vδ2+ γδ T cells and MAIT cells have been implicated in anti-viral immunity[14–16]. Activation of these cells by viruses is thought to occur via T-cell receptor (TCR) independent mechanisms, through IL-12 and IL-18 stimulation [14–16].

To examine the memory T cell response, PBMCs from the control infant and COVID-infected infant (day 84) were stimulated with inactivated SARS-CoV-2 (Fig. 3). A substantial increase in CD69 expression was observed in CD4+ (27.2%) and CD8+ (20.2%) T-cells compared to the age/sex-matched control (<5%) (Fig. 3A). This was accompanied by robust production of both pro- and anti-inflammatory cytokines and chemokines (IL-2, IL-4, IL-5, IL-6, IL-9, IL-10, IL-13, IL-15, IL-17, IFNγ, IP-10, TNFα) in PBMC supernatants, which was not observed in our control patient (Fig. 3B and Supplementary Fig. 2). These data reveal a strong memory T cell response to SARS-CoV-2, suggestive of long-lasting protective T cell-mediated immunity.

We have provided comprehensive longitudinal analyses of the clinical, immunological and virological findings in an infant with severe COVID-19. Overall, our cellular and cytokine findings support previous studies demonstrating depletion of lymphoid and innate cell populations in the early phase of severe COVID-19[17–19]. While a number of these observations have also been

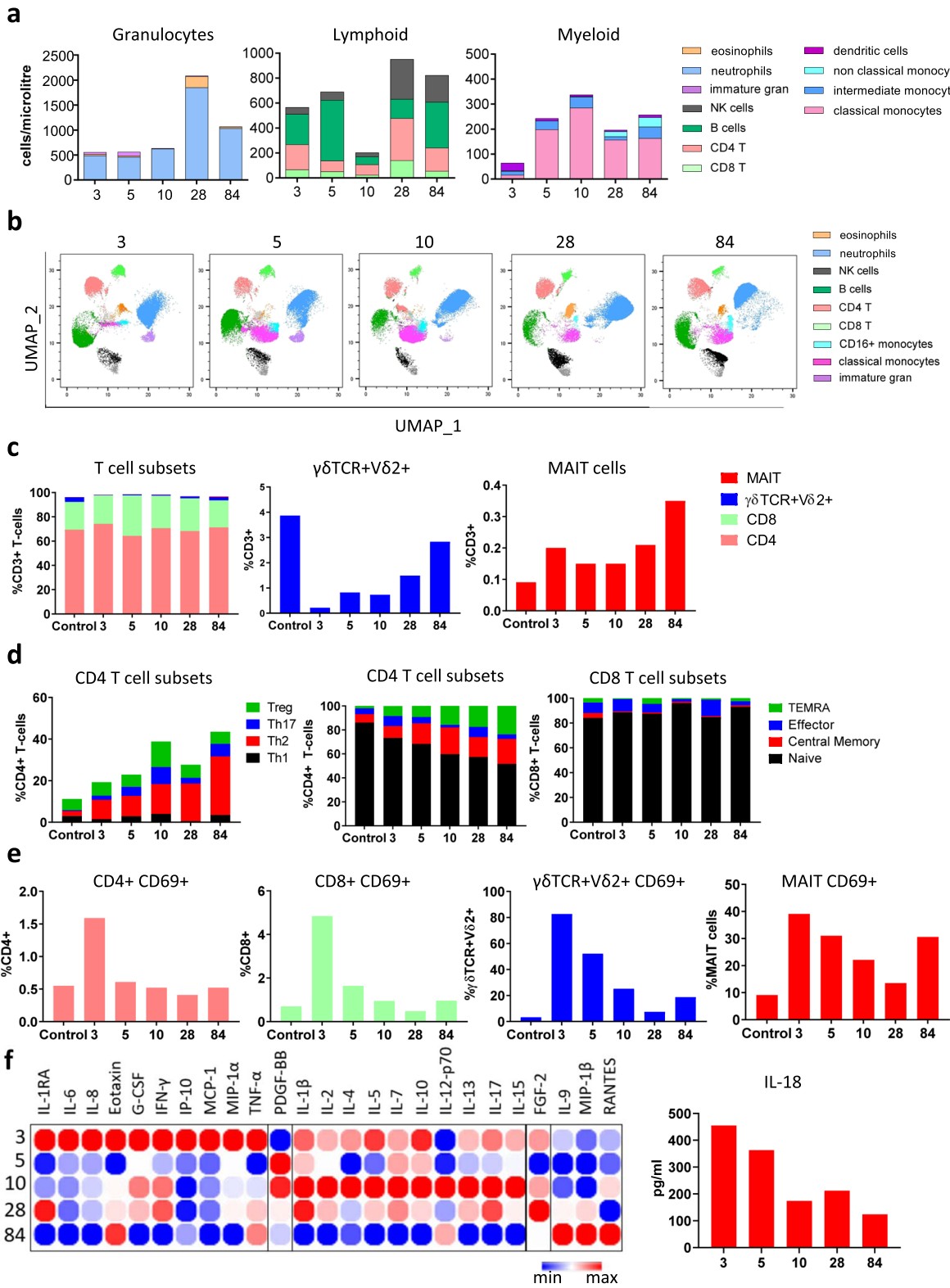

described in adults with severe disease (e.g., Th1 and Th17 CD4+ and CD8+ activation, eosinophilia and elevated pro-inflammatory cytokines in the serum such as IL-6, IFNγ, IP-10, and TNFα)[12, 20–24], we report several unique findings in this infant with severe COVID-19. First, the immune response was characterised by early and marked alterations in neutrophil and monocyte populations, expansion of CD4+ (but not CD8+) T cell populations coinciding with robust antibody responses, activation

of innate-like immune cells (e.g., CD69+ MAIT cells), and Th2 and IL17 cytokine skewing. Second, robust and enduring memory T cell responses were maintained for at least 84 days, suggesting functional immune memory. Third, activation of innate-like T cells was observed. This is consistent with studies in adults showing MAIT cell activation in severe COVID-19[25], suggesting it may also represent a biomarker for severe COVID-19 infection in children. Interestingly, the findings of elevated Th2 responses in

**Fig. 2 Immune cell profiling in whole blood and PBMCs. A** Flow cytometry was performed on whole blood samples collected at 3, 5, 10, 28, and 84 days following admission. Neutrophils, eosinophils, CD4 T cells, CD8 T cells, B cells, Natural Killer (NK) cells, monocytes (classical, intermediate, and non-classical), and dendritic cells were classified by manual gating and expressed as cells/µL using counting beads. **B** Unsupervised clustering and dimensionality reduction were performed using FlowSOM and UMAP on a concatenated file containing 150,000 live single cells (30,000 randomly selected cells from each time point). The UMAP plots at each time point are coloured according to the generated FlowSOM clusters. **C** Flow cytometry was performed on peripheral blood mononuclear cells collected on days 3, 5, 10, 28, and 84 following admission. CD4, CD8, γδVδ2+ T-cells, and MAIT cells were classified by manual gating and expressed as a frequency of CD3+ T-cells. These results were compared to an age/sex-matched control, obtained at a single time-point. **D** Subsets of CD4+ and CD8+ T-cells were further categorised by flow cytometry to determine the frequency of Th1, Th2, Th17, Treg, and memory subsets. These were expressed as the frequency of CD4+ or CD8+ T-cells. **E** Flow cytometry was performed to assess the activation status of T-cells. CD69+ was expressed as a proportion of total CD4+, CD8+, γδVδ2+ T-cells, and MAIT cells. **F** Cytokines were quantified in the serum using a multiplex cytokine assay and visualised in a heatmap containing log2 transformed values and clustered according to peak expression at day 3 (cluster 1), day 5 (cluster 2), day 10 (cluster 3), and day 28 (cluster 4). IL-18 levels at days 3, 5, 10, 28, and 84 were quantified by ELISA and expressed as pg/ml. UMAP Uniform Manifold Approximation and Projection, MAIT mucosal-associated invariant T cells.

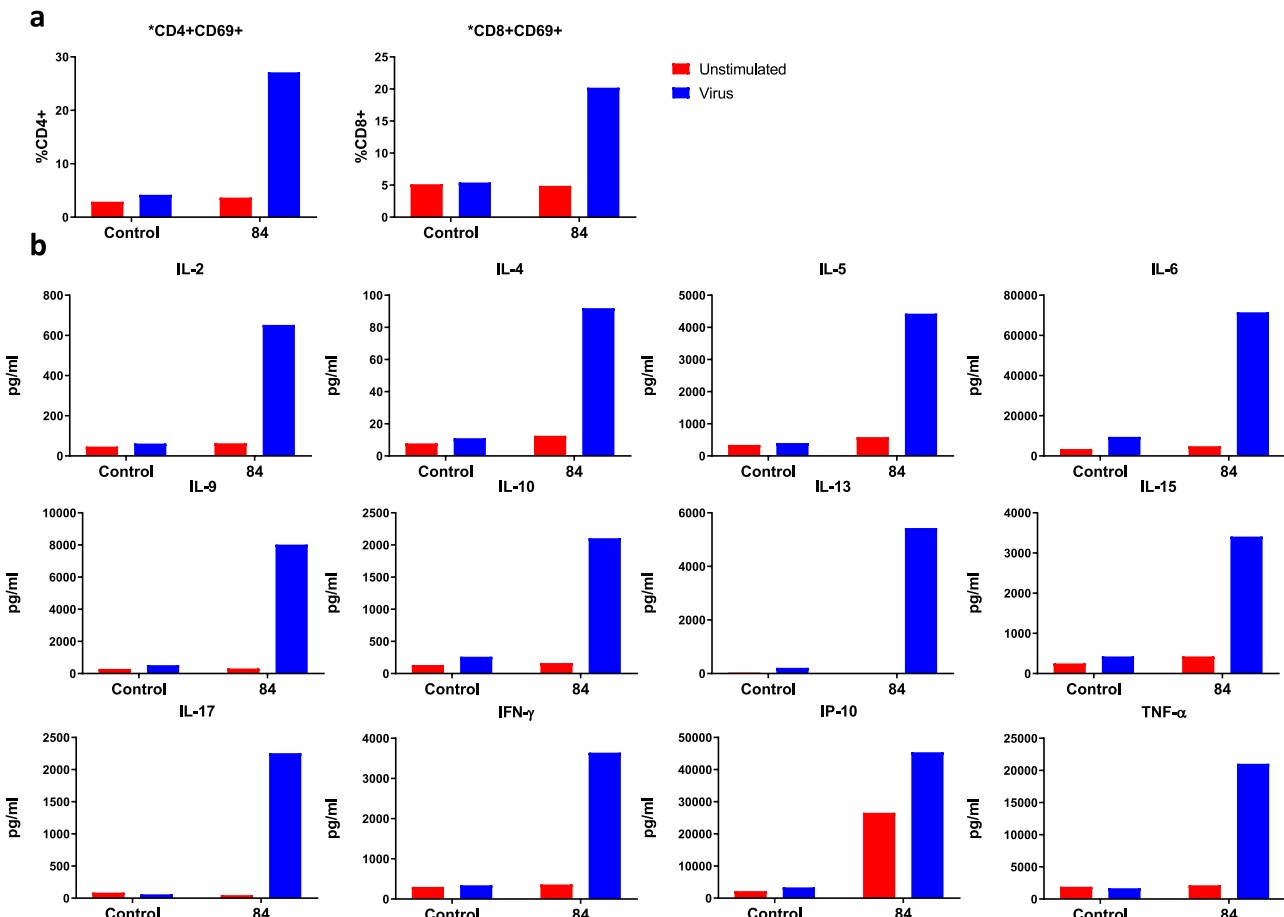

**Fig. 3 Immune response to inactivated SARS-CoV-2. A** Flow cytometry was performed on PBMCs following 4-day stimulation with inactivated SARS-CoV-2 to observe T-cell activation status. CD69+ was expressed as a proportion of total CD4+ or CD8+ T-cells. **B** Cytokines were quantified in PBMC supernatants following 4-day stimulation with inactivated SARS-CoV-2 by multiplex cytokine assay to assess memory T-cell responses.

our severe inpatient that persisted to day 84 contrasts with Jia et al's[26] findings which reported an early increase of Th2 cells in children during the acute phase of mild disease but resolved during convalescence. It is unclear whether Th2 responses in children are protective or contribute to disease severity, although it is possible that the patient's prior cardiac history also rendered the infant more susceptible to severe disease [27].

Overall, the findings support long-lived cellular immunity to SARS-CoV-2 infection in a child with complex cyanotic congenital heart disease, which was marked by an elevated inflammatory immune response in the acute phase of the infection, followed by Th2 skewing and prolonged T cell activation. This study provides insight into the immunological mechanisms underlying severe COVID-19 in an infant with congenital heart disease and may inform future research into potential therapeutic targets to mitigate severe disease in high-risk groups.

**Reporting summary.** Further information on research design is available in the Nature Research Reporting Summary linked to this article.

## Data availability

The datasets generated during and/or analysed during the current study are available from the corresponding author on reasonable request. Source data for results are provided in Supplementary Data 1.

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

## Acknowledgements

First, we wish to acknowledge this child's family for their generosity and willingness to contribute to research at a time of great stress. We also acknowledge the Clinical Laboratory team at The Royal Children's Hospital, Melbourne for their support. This study was funded by Murdoch Children's Research Institute COVID-19 research program; Centres of Excellence in Influenza Research and Surveillance—Cross-Centre Southern Hemisphere Project, National Institute of Health (NIH); The Influenza Complications Alert Network Surveillance System (FluCAN); Paediatric Active Enhanced Disease Surveillance (PAEDS) and Sentinel Travellers and Research Preparedness Platform for Emerging Infectious Disease (SETREP-ID)—Australian Partnership for Preparedness Research on Infectious Disease Emergencies. Funding was also provided through the Victoria Government's Operational Infrastructure Support Programme. P.V.L. is supported by Australian National Health and Medical Research Council (NHMRC) Career Development Fellowship. DGP is supported by a CSL Centenary Fellowship.

## Author contributions

D.W. co-conceptualised the study, contributed to analyses, and co-wrote the manuscript with M.R.N. and J.A. M.R.N., J.A., Z.T., R.H., C.D., L.A.H.D. planned, performed, and analysed the experiments with the assistance of P.V.L., D.G.P., J.B., S.J.; T.D, T.C., D.B., A.D., J.B., S.M., G.H., A.S., K.S., V.C., A.D., P.B., L.Y.L., Y.N.A., and S.T. provided clinical expertise and contributed to data interpretation. A.A., A.M., K.D., I.O., and I.K. collected and/or processed patient samples and assisted with manuscript preparation. P.V.L, D.G.P., and N.C. conceptualised the study and drafted the manuscript.

## Competing interests

The authors declare no competing interests.
