## [Peer Review File · Communications Medicine]

Reviewers' comments:

Reviewer #1 (Remarks to the Author):

This study provides a very detailed description of the clinical presentation and immune response development following SARS CoV-2 infection in a 4-month-old infant with history of congenital heart disease. The acute phase of infection was marked by elevated levels of pro-inflammatory cytokines, whereas a Th2 bias was observed at later time points. Virus-specific T cells responses were still detected 84 days after admission. Binding and neutralizing antibody responses were detected from 3-5 days post admission and persisted several weeks post infection (IgG was still detected 84 days post admission). The manuscript is well-written and the observations are interesting. However, it is unclear if the data observed in this case report based on a single infant are representative of severe SARS CoV-2 infection in infants and whether the data presented in this manuscript will be very informative for the field.

1-A throughout discussion of how the immune response in this infant compares to that of adults with severe infection and to that of other infected infants/children (both severe and mild infection) will be very helpful to fully interpret the observations in this infant.

2- The potential impact of the treatment (notably dexamethasone) on the kinetics of the cellular and humoral immune response is warrant a bite more discussion.

3- When did the infant start presenting clinical symptoms? It will be helpful to relate the immune measurements to clinical symptoms onset and not just hospital admission day.

4- Can the authors comment on the persistence of high proportion of Th2 cells at Day 84 post admission and on the fact that elevated Th2 responses have been thought to be associated with protection from severe disease in children? It is interesting that at Day 3, the proportion of Th2 cells in the severely infected infant is higher than in the control infant, despite the predominantly pro-inflammatory cytokine profile. A discussion of this would be helpful? How does the percentage compare to that of age matched healthy infants based on the literature?

5- The scale of figure 1D suggests that the infant had stronger neutralizing antibody responses than binding responses, which is probably not the case. It would be helpful to normalize the scale of the axis to that of a standard. Adjusting the scale of the binding response would notably allow to better appreciate the levels of IgG antibodies during the acute phase of infection.

6- It will be helpful to indicate on figure 1F when tocilizumab was administered. What does the dash lines represent in figure 1F?

7- In figure 2, were PBMC from the control tested at a single time point or were PBMCs tested over time? As the kinetics of immune cells significantly change early in life, it will be important to determine what changes are related to SARS CoV-2 infection versus immune maturation.

Reviewer #2 (Remarks to the Author):

The paper describes the case of a 4-month old child with a severe COVID-19. SARS-CoV-2 infection is very rare in newborns and infants .

The described child had a cyanotic congenital hearth disease partially corrected with a systemic-to pulmonary-shunt inserted at day 8 of life.

The infection had the typical course of the severe forms, with early mild symptoms followed by a rapid deterioration requiring treatment in the intensive care unit. Thanks to the implement of the appropriate antiviral and anti-inflammatory therapies the child recovered in a relatively short time.

The Authors describe the clinical and immunological findings in details.

They demonstrate that, although the early inflammatory reaction was similar to that observed in the adults, the child response to the virus was also characterized by the expansion of CD4, but not CD8 T cells and the timely production of neutralizing antibodies. The observation of the increase of MAIT cells may also important for their ability to protect mucosal sites.

I find the paper of interest to the community because of the new insights and indications on the immune response to SARS-CoV-2 of infants.

Minor observation

1. I have not found in ref.1 the data about the increase severity of COVID-19 in children younger than 4 years of age,
2. Probably the basic disease and the shunt favoured the triggering of the inflammatory reaction, very common in older adults and rare in children.

2 July 2021

Communications Medicine
Nature Research

Dear reviewers,

Thank you for your constructive comments and expert feedback which we believe have substantially enhanced the quality of our manuscript. Please find below our point-by-point responses, addressed with tracked changes in our manuscript.

Thank you once again.

Yours sincerely,

Danielle Wurzel (on behalf of co-authors)

Reviewer #1 (Remarks to the Author):

This study provides a very detailed description of the clinical presentation and immune response development following SARS CoV-2 infection in a 4-month-old infant with history of congenital heart disease. The acute phase of infection was marked by elevated levels of pro-inflammatory cytokines, whereas a Th2 bias was observed at later time points. Virus-specific T cells responses were still detected 84 days after admission. Binding and neutralizing antibody responses were detected from 3-5 days post admission and persisted several weeks post infection (IgG was still detected 84 days post admission). The manuscript is well-written and the observations are interesting. However, it is unclear if the data observed in this case report based on a single infant are representative of severe SARS CoV-2 infection in infants and whether the data presented in this manuscript will be very informative for the field.

1-A throughout discussion of how the immune response in this infant compares to that of adults with severe infection and to that of other infected infants/children (both severe and mild infection) will be very helpful to fully interpret the observations in this infant.

Response: Thank you for this suggestion. There are several studies in adults with severe COVID-19 disease but a notable paucity of studies in children with severe disease. Severe COVID-19 disease in adults results in lymphopenia, activation of CD4 (Th1 and Th17 mainly) and CD8 T cell populations, high levels of pro-inflammatory cytokines such as IL-6, TNF α and IL-1 β , activation of MAIT cells and potent NAb. In children with severe COVID-19, one study found increased levels of IL-17A and IFN γ , compared with adults, while other studies have reported severe COVID-19 disease to be associated with increased IP-10 levels and increased eosinophils. In our study, several features were in common with these findings, including robust

activation of CD4 and CD8 T cells, induction of strong NAb and elevated cytokines such as IFN γ , IL-6, IP-10 and TNF α during the acute phase. In contrast, our severe inpatient had an expanded Th2 cell population by day 3 which persisted until day 84 and was associated with Th2 cytokines in the serum by day 10. Eosinophilia was also observed by day 28. Therefore, while severe COVID-19 disease in this paediatric inpatient shared a number of features with severe disease in adults, there were also several unique differences in the immune response observed. We have added some further discussion of this in the revised manuscript (pages 4, para 1 and 2; and page 7, para 2).

2- The potential impact of the treatment (notably dexamethasone) on the kinetics of the cellular and humoral immune response is warrant a bite more discussion. comment in nature re dexamethasone study

Response: *Thank you for this suggestion. Dexamethasone is known to suppress the immune system with effects against a broad range of immune parameters, consistent with the observations in this study. We have included information related to the effect of dexamethasone in the revised manuscript (page 6, para 1).*

3- When did the infant start presenting clinical symptoms? It will be helpful to relate the immune measurements to clinical symptoms onset and not just hospital admission day.

Response: *A day prior to presentation this infant developed low-grade fevers, cough and increased work of breathing. The following day his respiratory status worsened and he developed severe respiratory and metabolic acidosis with pH 6.99, PCO₂ 55 mmHg and lactate 13 mmol/L. He was intubated and ventilated and transferred to the intensive care unit for further management. Hence day 0 in our timeline for Figure 1 refers to the day of admission to our hospital, which was day 2 of the child's illness. We have revised this accordingly to indicate when clinical symptoms developed in this patient (page 2 of the revised manuscript; and modified Figure 1A).*

4- Can the authors comment on the persistence of high proportion of Th2 cells at Day 84 post admission and on the fact that elevated Th2 responses have been thought to be associated with protection from severe disease in children? It is interesting that at Day 3, the proportion of Th2 cells in the severely infected infant is higher than in the control infant, despite the predominantly pro-inflammatory cytokine profile. A discussion of this would be helpful? How does the percentage compare to that of age matched healthy infants based on the literature?

Response: *The role of Th2 cells in protecting children against severe COVID-19 is not fully understood. Some studies suggest that Th2 responses might be protective in children, given the Th2 bias normally seen during early life but there have not been many studies investigating this to date. Whether this Th2 response could protect against re-infection is a key question. In adults,*

Th2 responses appear to correlate with more severe disease outcomes. We have added some discussion of this in the revised manuscript (pages 7-8).

The percentage of Th2 cells seen in our healthy age-matched control is similar to that observed in other studies (around 2-3%). We have added a reference for this (Botafogo et al, Front Immunol 2020) on page 5 of the revised manuscript

5- The scale of figure 1D suggests that the infant had stronger neutralizing antibody responses than binding responses, which is probably not the case. It would be helpful to normalize the scale of the axis to that of a standard. Adjusting the scale of the binding response would notably allow to better appreciate the levels of IgG antibodies during the acute phase of infection.

Response: We have modified Figure 1D as requested.

6- It will be helpful to indicate on figure 1F when tocilizumab was administered. What does the dash lines represent in figure 1F?

Response: We have modified Figure 1F as requested.

7- In figure 2, were PBMC from the control tested at a single time point or were PBMCs tested over time? As the kinetics of immune cells significantly change early in life, it will be important to determine what changes are related to SARS CoV-2 infection versus immune maturation.

Response: We thank the reviewer for the suggestion and agree the immune system can significantly change early in life. The control data we provide is from an age/sex matched PBMCs from a single time-point. Unfortunately, we cannot access further PBMCs from the control donor to match the 3 month time-point of the inpatient.

Reviewer #2 (Remarks to the Author):

The paper describes the case of a 4-month old child with a severe COVID-19. SARS-CoV-2 infection is very rare in newborns and infants .

The described child had a cyanotic congenital hearth disease partially corrected with a systemic-to pulmonary-shunt inserted at day 8 of life.

The infection had the typical course of the severe forms, with early mild symptoms followed by a rapid deterioration requiring treatment in the intensive care unit. Thanks to the implement of the appropriate antiviral and anti-inflammatory therapies the child recovered in a relatively short time.

The Authors describe the clinical and immunological findings in details.

They demonstrate that, although the early inflammatory reaction was similar to that observed in the adults, the child response to the virus was also characterized by the expansion of CD4, but

not CD8 T cells and the timely production of neutralizing antibodies. The observation of the increase of MAIT cells may also be important for their ability to protect mucosal sites.

I find the paper of interest to the community because of the new insights and indications on the immune response to SARS-CoV-2 of infants.

Minor observation

1. I have not found in ref.1 the data about the increase in severity of COVID-19 in children younger than 4 years of age

Response: Thank you for this comment. In Table 1 of reference 1. (Leidman, MMRW 2021), the third age bracket (0-4 years) shows the highest rates of hospitalisation and intensive care unit admission of any other age brackets in 0<24 year old age range. We have added this additional detail in the Summary (Page 2, lines 3-4 of the summary).

Of note, a typo was noted for the child's age, now corrected as 5 months. The control was aged and sex matched as described.

2. Probably the basic disease and the shunt favoured the triggering of the inflammatory reaction, very common in older adults and rare in children.

Response: We completely agree that the patient's previous medical history likely contributed to the severity of COVID-19 disease consistent with a recent systematic review that showed that pre-existing cardiac disease is the most common comorbidity amongst children requiring ventilation for SARS-CoV-2 infection (Williams, Eur J Ped, 2021). We have included additional information about the patient's heart condition on page 2 (para 2) of the revised manuscript and also on page 8 (para 2).

Thank you once again to the expert reviewers for their feedback and suggestions and to the editors for considering our revised manuscript.

Yours sincerely,

Dr Danielle Wurzel (on behalf of the study authors)

REVIEWERS' COMMENTS:

Reviewer #1 (Remarks to the Author):

This is a significantly improved resubmitted manuscript in which the authors report the clinical presentation and immune response to SARS CoV-2 infection in a 5 month-old infant with history of congenital heart disease. Similar to responses reported in adults with severe SARS CoV-2 infection, the acute phase of infection was marked by elevated levels of pro-inflammatory cytokines and by a depletion of lymphoid and innate cells. But the authors also noted some responses that were distinct from the response reported in adults including Th2 responses that were still detected 84 days after admission. Binding and neutralizing antibody responses were also durable with S1-specific IgG still detected 84 days post admission. It is possible that the observed immune responses were impacted by the treatment given including dexamethasone, and that the severity of infection was related to the prior medical history of the patient as discussed in the paper. Nevertheless, the importance of this study relays on the fact that while several studies have characterized responses in adults with severe SARS CoV-2 infection, a few reports on pediatric severe SARS CoV-2 infection, especially amount infants have been published.

The revised manuscript addresses all major points noted in the initial review and we have no major comment. We do appreciate the fact that the authors added in the title that this is a case report as one minor concern remains whether the findings in this one case with prior history of cardiac diseases reflect the clinical and immunological presentation of severe SARS CoV-2 infection in infants.

Another minor concern is with the control in figure 2. While we appreciate that longitudinal samples could not be collected from the control patient, we feel that this needs to be better clarify in the text instead of the use of "age/sex matched control".

Overall, we think that well detailed case reports like the one presented in this study will allow for a better understanding of severe SARS CoV-2 infection in young children.